# Properties and Fabrication of Waterborne Polyurethane Superhydrophobic Conductive Composites with Coupling Agent-Modified Fillers

**DOI:** 10.3390/polym14153093

**Published:** 2022-07-29

**Authors:** Fangfang Wang, Jihao Ci, Jiang Fan

**Affiliations:** 1Material Engineering School, Shaanxi Polytechnic Institute, Xianyang 712000, China; 2College of Energy Engineering, Xi’an University of Science and Technology, Xi’an 710054, China; 3College of Geology and Environment, Xi’an University of Science and Technology, Xi’an 710054, China; wangfangfang410@163.com; 4Department of Chemical Engineering, Textile and Clothing, Shaanxi Polytechnic Institute, Xianyang 712000, China; fanj1991@126.com

**Keywords:** multi-walled carbon nanotubes, polytetrafluoroethylene, electrostatic spraying, waterborne polyurethane, superhydrophobic conductive coatings, coupling agent

## Abstract

The addition of abundant fillers to obtain conductive and superhydrophobic waterborne polyurethane (WPU) composites generally results in increased interfaces in the composites, leading to reduced adhesion and poor corrosion resistance. Fillers such as Polytetrafluoroethylene (PTFE) and multi-walled carbon nanotubes (MWCNTs) were first treated by a coupling agent to reduce the contents of the fillers. Thus, in this work, WPU superhydrophobic conductive composites were prepared using electrostatic spraying (EsS). The polar groups (-OH and -COOH, etc.) on the WPU, PTFE, and MWCNTs were reacted with the coupling agent, making the WPU, PTFE, and MWCNTs become crosslinked together. Thus, the uniformity of the coating was improved and its curing interfaces were reduced, causing enhanced corrosion resistance. The dehydration reaction that occurred between the silane coupling agent and the polar surface of Fe formed -NH_2_ groups, increasing the adhesion of the coating to the steel substrate and then solving the problems of low adhesion, easy delamination, and exfoliation. With the increased content of the modified fillers, the conductivity and hydrophobic property of the composite were amplified, and its corrosion resistance and adhesion were first strengthened and then declined. The composite with the WPU, PTFE, MWCNTs, and KH-550 at a mass ratio of 7:1.5:0.1:0.032 held excellent properties; its volume resistivity and WCA were 1.5 × 10^4^ Ω·cm and 155°, respectively. Compared with the pure WPU coating, its adhesive and anticorrosive properties were both better. This provides a foundation for the fabrication and application of anticorrosive and conductive waterborne composites.

## 1. Introduction

To avoid catastrophic accidents such as fire disasters caused by electrostatic sparks arising from chemical raw materials and oil products, conductive coatings are widely used to protect the external surfaces of the equipment in the petrochemical industry [1,2]. Filling-type conductive coatings are mainly composed of polymers, conductive fillers, and solvents, which have shown excellent comprehensive performance in practical application and have become the research focus of the conductive coating industry [3]. Owing to the use of water as the disperse medium, waterborne polyurethane (WPU) coatings with excellent properties of PU resins are a class of reliable and eco-friendly coatings [4,5]. However, the poor water contact angle (WCA) and the low waterproof property cause traditional WPU coatings to have restricted application in the field of anticorrosive conductive coatings. Previous studies have shown that superhydrophobic coatings can be constructed with two factors: low-surface energy materials and rough surfaces [6].

Owing to a hollow tubular structure and a certain aspect ratio, the addition of a few carbon nanotubes (CNTs) to the resins can form a network structure [7,8]. Polytetrafluoroethylene (PTFE) materials with high fluoride content hold properties of high hydrophobicity and low friction. Thus, the addition of an appropriate amount of PTFE as a hydrophobic filler will cause a reduced surface free energy of the WPU coating [9,10]. Generally, it is necessary to add a large number of CNTs and PTFE materials to obtain conductive and superhydrophobic WPU coatings [11], but the addition of an excessive amount of these fillers leads to increased coating interfaces, reduced adhesion, poor corrosion resistance, and other properties [12]. In particular, the addition of an excessive amount of PTFE materials results in an increased resistance of the coatings and makes them susceptible to peeling off from the metal substrates.

Coupling agents as a kind of additive for polymer composites contain two functional groups with different chemical properties at the molecular level. Under the action of the conjugation arising from the coupling agents, the chemical reaction that occurs between the inorganics and organics forms a “bridge” in the inorganic–organic interfaces, leading to close integration between them [13]. Thus, the compatibility and the interface bond between the inorganic–organic phase are improved. The KH-550 silane coupling agent, which can react with the surfaces of substances with polar groups, was used to modify the fillers in this work to prepare conductive and superhydrophobic coatings with low filler content [14,15]. The dehydration reaction that occurred between the coupling agent and the surface of Fe formed the -NH_2_ groups and increased the adhesion of the coating to the Q235 steel substrate [16]. Moreover, the dehydration reaction that occurred between the coupling agent and the organics such as WPU resins improved the compatibility of the WPU and the PTFE.

For electrostatic spraying (EsS), a jet with a spinous edge in the spray gun head immediately generates high-voltage discharges and air ionization after the connection of the high-voltage negative electricity [17]. Under electrostatic attraction, the liquid dispersions holding negative charges sprayed from the nozzle are coated on the steel substrates holding positive charges. Additionally, the impact force from the compressed air acts as a tractive force to effectively atomize the dispersions. It can prevent the conductive fillers from undergoing agglomeration to some extent and improve the uniformity of the coatings by controlling the thickness and shortening the curing time [18]. In this way, a process of EsS with good capability of atomization was used for spraying in this work.

Based on the practical problems above, the MWCNTs and PTFE were first modified by a kind of coupling agent and then added to the WPU dispersions to prepare the mixed WPU dispersion system. Subsequently, different mixed WPU dispersions were respectively sprayed on the metal substrates by EsS. Thus, the WPU superhydrophobic conductive composites (WPU SCCs) with coupling agent-modified fillers were obtained under the synergistic effect between the low surface energy from the PTFE and the micro-rough surface structure from the MWCNTs. The dispersion of the MWCNTs modified by the coupling agent and the effect of the modified fillers (M-MWCNTs-PTFE) on the performances of the coating were respectively studied. This provides an important foundation for the preparation of anticorrosive conductive waterborne composites with high adhesion and self-cleaning properties using EsS.

## 2. Materials and Methods

### 2.1. Experimental Materials

The WPU was purchased from Jining Huakai Resin Co. Ltd. (Jining, China). Its V.O.C content, solid content, specific gravity, and viscosity were 253 (g·L^−1^), 35%, 1.054 (g·mL^−1^), and 75 cps, respectively. The PTFE resins was purchased from Ricaron Polymer Materials Co., Ltd. (Dongguan, China), oyster white. The MWCNTs particles (FloTube 9000 series) were purchased from Beijing Tiannai Technology Co., Ltd. (Beijing, China). The average diameter, purity, average length, and density were 10~15 nm, 95%~97.5%, 10 μm, and 0.03~0.15 (g·cm^−3^), respectively. 3-aminopropyl triethoxysilane (KH-550 coupling agent, chemically pure) was purchased from Shanghai McLean Biological Co., Ltd. (Shanghai, China). Sodium chloride (NaCl, analytical pure) was purchased from Tianjin Komil Chemical Reagent Co., Ltd. (Tianjin, China).

In accordance with ISO 8501-1, the Q235 steel substrates (50 × 25 × 3 mm) were sandblasted by a YX-6050A sand blasting device (Anbangruiyuxin Machine Technology Development Co., Ltd., Wuhan, China); its air pressure, spray distance, and spray time were 0.6~0.8 MPa, 110~150 mm, and 40~50 s, respectively. Subsequently, they were ultrasonically cleaned in anhydrous ethanol and then kept at room temperature [19]. Thus, the surface state and the roughness level were Sa 2.5 and G, respectively, in accordance with ISO 8503-1.

### 2.2. Preparation of the WPU Dispersions with Modified Fillers

The KH-550 coupling agent was blended with the distilled water at a mass ratio of 1:1 using 85–2 magnetic stirring equipment (Hangzhou Instrument Motor Co., Ltd., Hangzhou, China) to hydrolyze at room temperature. After that, the hydrolytic coupling agent was stored for use at room temperature.

Fillers of the MWCNTs, PTFE emulsion, and hydrolyzed KH-550 coupling agent (named as M-MWCNTs-PTFE fillers) were mixed with the WPU at 100~150 r·min^−1^ for 30 min using 85–2 magnetic stirring equipment and then treated for 5 min with KQ-50B ultrasonic dispersion equipment (Kunshan Ultrasonic Instrument Co., Ltd., Kunshan, China) to obtain the mixed WPU dispersions with modified fillers (M-MWCNTs-PTFE/WPU). Then, they were stored at room temperature for 20~30 min. For #1, #2, #3, and #4 (shown in Table 1), the contents of the MWCNTs were 0.4 wt%, 0.8 wt%, 1.2 wt%, and 1.6 wt%, respectively, and the contents of the PTFE emulsions were 13.0 wt%, 16.5 wt%, 17.4 wt%, and 21.9 wt%, respectively.

### 2.3. Preparation of the Coatings

The WPU SCCs were obtained by spraying the M-MWCNTs-PTFE/WPU dispersions on the steel substrates using NEW KCI-CU801 electrostatic spraying equipment (Shenzhen Honghaida Instrument Co., Ltd., Shenzhen, China). The process parameters of the sandblasting treatment are shown in Table 2. During spray, the coating thickness was controlled at 80~90 μm by weighting. Then, the samples were cured for 1 day at room temperature and then for 1 h at 150 °C in the oven. They were named as #0 (pure WPU coating), #1, #2, #3, and #4 in accordance with Table 1. 

### 2.4. Characterization and Test

#### 2.4.1. Electrical Conductivity

According to GB/T 16906-1997, the surface resistance of the composite was tested using a KEYSIGHT B2985A high resistance meter (Agilent, America) and silver glue platinum wires at 100 V. Three samples for each coating and three points for each sample were measured. The thickness of the coating was tested using an HCC-18 magnetoresistive thickness meter (Shanghai Huayang Testing Instrument Co., Ltd., Shanghai, China). Five samples for each coating and six locations for each sample were conducted. The average volume resistivity was evaluated by the thickness and the surface resistance. The dispersibility of the MWCNTs was observed using a VEGA3 XMU Scanning Electron Microscope (SEM) (TESCANSCAN, Brno, Czech).

#### 2.4.2. Hydrophobicity Test

The water contact angles (WCAs) of the coatings were measured using an SDC-200 contact angle testing machine (Dongguan Shengding Precision Instrument Co., Ltd., Guangzhou, China). A pure water droplet of 20 μL was used. Three samples for each coating were measured, six times for each sample. The 3D-micro surface morphologies of the coatings were observed using an LEXT OLS4000 laser confocal scanning microscope (OLYMPUS, Tokyo, Japan) at room temperature; the field of view was 16 μm–2.56 mm and the magnification was 108–17,280×.

#### 2.4.3. Adhesion Test

At room temperature, the pull-off test was conducted using an HT-2402 computer servo control materials testing system (Hung Ta Instrument, Co., Ltd., Taiwan). According to ISO 4624:2002, the tested coatings were prepared and the samples were bonded by the cyanoacrylate adhesive. The tensile stress was conducted at an evenly increasing rate in the direction perpendicular to the plane of the coating. Five samples for each coating were tested to calculate the average adhesion. The wear tracks were measured using a VEGA3 XMU Scanning Electron Microscope (SEM) (TESCANSCAN, Brno, Czech).

#### 2.4.4. Anticorrosion Property Test

The anticorrosion property of the coating was evaluated through electrochemical impedance spectroscopy (EIS) and a polarization curve, which were performed with a Ver4.2corr test system using a CS350H Electrochemical Workstation (Wuhan Corr Test Co., Ltd., Wuhan, China) with a three-electrode cell at room temperature. The saturated calomel electrode served as the reference electrode, the Pt electrode served as the auxiliary electrode, and the coating with an exposed area of 0.785 cm^2^ served as the working electrode. The samples were immersed in 3.5 wt% NaCl solution at 40 °C for 30 days and the solution was changed every two days before the test. The EIS was tested after the open circuit potential of the system was steady. The sine wave amplitude was 20 mV and the frequency was 10^−2^~10^5^ Hz. During the test of the polarization curve, the scanning speed was 0.2 mV·s^−1^, and the initial potential and the final potential were −0.2 V and 0.3 V, respectively.

## 3. Results and Discussion

### 3.1. Electrical Conductivity

Table 3 shows the volume resistivities of the WPU SCCs with different M-MWCNTs-PTFE contents. The electrical conductivity of the prepared composites was greatly influenced by the contents of the MWCNTs [20]. In addition, the dispersion of the modified MWCNTs was improved and thus the conductive network was formed as the MWCNT content was low [21,22]. Thus, the volume resistivity was mildly reduced (in the magnitude of 10^4^ Ω·cm) with the rise of the MWCNT content, arising in an increased conductivity. The conductive properties of the WPU composites with the fillers modified by the coupling agent (Table 3) were excellent, meeting the industry standard.

The appearance shown in Table 3 can be explained by the dispersion of the MWCNTs in the composites. The surface micromorphologies of #1 (Figure 1a), #2 (Figure 1b), #3 (Figure 1c), and #4 (Figure 1d) under a magnification of 50.00 k× and that of 0# (Figure 1e) under a magnification of 2.00 k× were observed, as shown in Figure 1.

As shown in Figure 1a,b, the distribution of the modified MWCNTs in #1 and #2 was relatively uniform and there were few apparently agglomerated MWCNTs. It may be that the compatibility between the MWCNT and the resin was improved under the chemical reaction between them and the KH-550 coupling agent. Due to the thinner polymer isolation layer between the MWCNT and the MWCNT, a tunnel conductive effect was generated in the composite. As a result, its volume resistivity was high and its electrical conductivity was weak. The MWCNT content of #3 (Figure 1c) and that of #4 (Figure 1d) were significantly more than that of #1, but the MWCNTs of them were still evenly dispersed without agglomeration. The reason was that the MWCNT content of #3 was smaller than #4 and the production of the conductive network among MWCNTs in #3 was weaker, leading to its low volume resistivity. It can be seen from Figure 1 that a dense and even grid structure was formed in the coating with a small amount of the MWCNTs, which provides a basis for the construct of the micro-rough surface structure on the waterborne composite.

### 3.2. Hydrophobicity

The WCAs of the WPU SCCs with different contents of the modified fillers are shown in Figure 2. The surface energy of the composite was greatly influenced by the PTFE content. Thus, the WCAs of the composites were progressively increased with increased PTFE content, leading to an obvious improvement of the hydrophobicity. The WCA of #0 was 84°, as shown in Figure 2. After the addition of the M-MWCNT-PTFE fillers, the WCAs of #1, #2, #3, and #4 were 123°, 132°, 155°, and 157°, respectively, indicating that the WPU SCCs were experimentally prepared. This may be explained by the following. The improved compatibility among the MWCNTs particles, the WPU resins, and the PTFE resins under the coupling effect of the coupling agent led to the PTFE being evenly dispersed in the top area of the coating, and then its surface energy was significantly decreased. The WCA of the composite with a PTFE content below 17.4 wt% was greatly strengthened with increased PTFE content. For #1 and #2, their WCAs were both far over 90°, demonstrating that they were significantly hydrophobic. It may be that the hydrophobicity of the composite with a low MWCNT content was greatly influenced by the PTFE content. During the preparation of the M-MWCNTs-PTFE/WPU dispersions, the polar groups on the surfaces of the MWCNTs, the PTFE resins, and the WPU resins interacted well with the coupling agent under the action of the magnetic agitation and the ultrasonic treatment. Then, the PTFE particles in the top area of the WPU coating were evenly adsorbed at the active points on the surfaces of the MWCNTs, causing a decreased surface energy of the composite and giving it good hydrophobicity. 

The hydrophobic groups on the surface of #3 holding 17.4 wt% PTFE and 1.2 wt% MWCNTs were under saturation conditions; the WCA of #3 was more than 150° and it became superhydrophobic. The reason may be explained as follows. Firstly, the bonding state of the MWCNTs particles to the resins was improved after the treatment of KH-550 coupling agent, and then the stability of the modified WPU dispersions was better. Secondly, the MWCNTs were first entered into the area of the coating with fewer MWCNTs under the electrostatic force and the impact force of the compressed air during spray. Due to the high MWCNT content of #3, the MWCNTs in the top area of the composite were evenly dispersed, and then the formed micro-rough surface structure was more uniform and compact. In addition, the surfaces of the MWCNTs in the top area were attached to a uniform layer of low-surface energy material due to the increased PTFE content, leading to the even hydrophobicity of the surface of #3. As the PTFE content was over 21.9 wt%, the change of the micro-rough surface structure was only partly due to the saturated MWCNT content in the top area. Finally, the WCA of the composite was slightly changed as the PTFE content was continuously increased.

The surface 3D-micro morphologies of the coatings (shown in Figure 3) were examined to further analyze the relationship between the surface roughness and the hydrophobicity. The average values of the surface roughness for different coatings are summarized in Table 4. It was shown that the surface state of #0 (Figure 3a) was clean and smooth, and the surface state of 1# (Figure 3b) was similar to 0#. However, the surface of #1 was mildly rougher than that of #0 due to the addition of the M-MWCNT-PTFE, indicating that the surface morphology of the composite only differed slightly, as the content of the MWCNTs modified by the coupling agent was lower than or equal to 0.4 wt%. With the improved dispersion of the modified MWCNTs, the obvious appearance of the micro-rough surface structures of #2 (Figure 3c), #3 (Figure 3d), and #4 (Figure 3e) with an increase in the MWCNT content from 0.8 wt% to 1.6 wt% can provide a foundation for the fabrication of the superhydrophobic composites. However, the content of the MWCNTs particles in the top area of #2 was small due to the low content of the modified MWCNTs, so that the upper layer of #2 was not entirely covered, and the micro-rough surface structure was not uniform or compact, though the coating showed good hydrophobic performance. Moreover, as the content of the modified MWCNTs of #3 was lower than that of #4, the MWCNTs particles inset in the upper layer of #3 were much more uniformly distributed, and its micro-rough surface structure became denser. This indicated that the hydrophobic property of the composite was in close relation with the micro-rough surface structure [23]. In addition, it indicated that the MWCNTs first modified by a small amount of coupling agent and then added into the WPU dispersions, subsequently sprayed using EsS, can structure a micro-rough surface structure on the composite, causing the foundation for the preparation of superhydrophobic coatings.

In accordance with Wenzel equation, $\cos\theta_{\omega }=r\cos\theta_{0}$, where $\theta_{\omega }$ is the apparent contact angle, $\theta_{0}$ is the eigen contact angle, and r is the micro-roughness. When $\theta_{0}$ was more than 90°, the hydrophobicity of the coating rose with an increase in r. With the addition of the modified fillers into the WPU, the WCAs of the WPU composites produced using EsS were over 90°, indicating that they became hydrophobic. Obviously, the micro-rough surface structure of the composite was rough as the content of the modified MWCNTs was only 0.8 wt%. Subsequently, the hydrophobicity of the composite was improved with an increase in the content of the modified MWCNTs. The WCA of the composite with 1.2 wt% MWCNTs was greater than 150°, and then a superhydrophobic coating was generated. 

According to the Cassie–Baxter equation, $\cos\theta_{c}=f(1+\cos\theta_{0})-1$, where $\theta_{c}$ is the apparent contact angle, $\theta_{0}$ is the eigen contact angle, and f is the solid–liquid contact area percentage. $\theta_{c}$ strengthened as f diminished. The $\theta_{0}$ and $\theta_{c}$ values of #3 were 84° and 155°, respectively, and it was calculated that its f value was 0.08; in other words, the gas–solid contact area was 92%, showing the existence of much gas between the liquid drop and the micro-rough structure of the composite surface. Thus, #3 became superhydrophobic and was used as a self-cleaning coating.

### 3.3. Adhesion to the Steel Substrate

Figure 4 shows the adhesion of different WPU SCCs with modified fillers to the Q235 steel substrates. It was shown that the adhesive property of #0 to the steel substrate was 5.72 MPa. The adhesive property between the composite and the steel substrate first strengthened and then diminished with increased PTFE content. According to ISO 12944-6, the adhesion of #1, #2, #3, and #4 conformed to the requirements for protective coatings (>5 MPa). Apparently, it was concluded that the adhesion of the composite to the steel substrate was greatly influenced by the content and the dispersibility of the MWCNTs as well as the PTFE. The appearance may be explained by the following. Because of the low content of the modified MWCNTs, the agglomerated MWCNTs particles after mechanical dispersion were further dispersed during spray. In addition, the polymerization reaction that was generated between the active groups on the surfaces of the MWCNTs modified by the silane coupling agent and the active groups on the WPU molecule generated a more uniform and firm network structure, so that the deposition of the MWCNTs was effectively reduced and the interface contact area of the resin/steel substrate was not affected. Moreover, the dehydration reaction that occurred between the silane coupling agent and the polar surface of Fe formed the -NH_2_ groups. Furthermore, due to the ability of the reaction between the silane coupling agent and the non-polar PTFE resins, the surfaces of the WPU and the PTFE were wrapped with a layer of silane coupling agent. Thus, the adhesive property of the composite to the Q235 steel substrate was strengthened. The surfaces of the MWCNTs, PTFE, and WPU were modified by the silane coupling agent to increase the surface activity and improve the solubility, resulting in them becoming crosslinked together. Finally, the anticorrosive property of the composite was enhanced and its internal delamination was also prevented.

As the PTFE content was below 17.4 wt%, the adhesion between the composite and the steel substrate was better than that of the pure WPU coating. As the PTFE content increased to 17.4 wt% and the MWCNT content was 1.2 wt%, the adhesion between #3 and the steel substrate was the highest among all the coatings, which was enhanced by 7.87% in comparison with #0. It may be that the M-MWCNT-PTFE content of #3 was at a critical value; the pores arising from the water evaporation in the WPU during curing were basically supplemented by the fillers. Furthermore, the adhesion of #3 (with the densest structure) to the steel substrate was the maximum among the whole composite. As the M-MWCNT-PTFE content was continuously increased, the MWCNT content as well as the PTFE content was so high that the coupling agent content was insufficient, resulting in the decreased adhesion of the coating. Thus, the adhesion between #4 and the steel substrate was significantly lower than the pure WPU coating.

### 3.4. Anticorrosive Property

The equivalent electric circuit models with two time constants of the pure WPU coating (Figure 5a) and the WPU SCCs with fillers modified by the coupling agent (Figure 5b) after immersion in NaCl solution were used to examine the impedance spectra. As stated in Figure 5, Rs is the electrolyte resistance, C is the interface double-layer capacitance of the coating, Rp1 is the effective path resistance, CPE is the double-layer capacitance, Rp2 is the charge diffusion resistance, and Wo is the Warburg impedance. The corrosive medium permeated into the Q235 steel immersed in NaCl solution for 30 days.

Figure 6 shows the complex plane impedance spectra (Figure 6a) and Bode plots (|*Z*|-*f* in Figure 6b and *θ*-*f* in Figure 6c) for different WPU SCCs with fillers modified by the coupling agent immersed in NaCl solution. It was concluded from Figure 6a,b that the impedance value of the composite was first improved and then reduced with increased PTFE content, demonstrating that the anticorrosive property first rose and then diminished. By the comparison of |*Z*|_0.01Hz_ values of different coatings in Figure 6, the impedance of the WPU SCCs with modified fillers by the coupling agent were at least two orders of magnitude better than the pure Q235 steel substrate, indicating that the composites—with their excellent property of acting as a barrier to the corrosive medium—can serve as anticorrosive materials for carbon steel. Two time constants were shown in the phase angle-frequency (Figure 6c). The high frequency region represented the reaction at the electrolyte–composite interface as well as the description of the shielding property of the composite. The low frequency range was characterized the corrosive process at the electrolyte/steel substrate interface as well as the penetration of the electrolyte solution into the coating. This indicated that the impedance of the WPU SCCs with modified fillers was more than the pure WPU coatings, indicating the good barrier effect of the prepared SCCs on the external corrosive media. This phenomenon may be explained as follows. The uniform dispersion of the MWCNTs and the PTFE under the electric field force during spray caused a diminished amount of microscopic defects in the composite. Moreover, the interfacial compatibility of the WPU, the PTFE, and the MWCNTs increased under the coupling effect of the coupling agent. Thus, the interface between the WPU and the PTFE was reduced, leading to a more compact coating structure. As the PTFE content was 17.4 wt% and the MWCNT content was 1.2 wt%, the impedance value of #3 was the best among all the coatings due to its even and compact micro-rough surface structure and excellent hydrophobicity property, which was about three orders of magnitude better than #0 and four orders of magnitude higher than the pure steel substrate. This indicated that the dense structure and good hydrophobicity of the coating arose in the excellent corrosion resistance. For #4, its high PTFE content and insufficient silane coupling agent content led to the existence of a certain interface between the WPU and the PTFE, resulting in decreased corrosion resistance. Meanwhile, the addition of an appropriate content of the silane coupling agent was a critical factor for the preparation of conductive composites with high corrosion resistance, wear resistance, and adhesion.

Figure 7 shows the potentiodynamic polarization curves of the WPU SCCs with modified fillers immersed in NaCl solution. Table 5 shows the fitted parameters of the potentiodynamic polarization curves. It was shown that the anticorrosive property of the WPU SCCs immersed in NaCl solution was significantly better than the pure WPU coating; their corrosion potentials were greater but their corrosion current densities were smaller than the pure steel substrate, showing that the WPU SCCs with fillers modified by the coupling agent prepared using EsS can give protection to the carbon steel. Furthermore, the reduced corrosion potential and the improved corrosion current density of the composite with an increased PTFE content indicated that the anticorrosive property first improved and then reduced. Remarkably, the anticorrosive property of #3 with 17.4 wt% PTFE and 1.2 wt% MWCNTs was the best—about four orders of magnitude smaller than the pure Q235 steel and about three orders of magnitude smaller than #0. This may be explained as follows. Both the MWCNTs and the PTFE, holding excellent anticorrosive properties, were well distributed under the coupling action and the electrostatic force during spray. In addition, the WPU and the PTFE were crosslinked together under the chemical reaction that occurred between them and the coupling agent, and then a coating basically without internal interfaces was formed, leading to a reduction in the microdefects and diffusion route of the electrolyte in the composite. Thus, the shielding effect of the composite coating was enhanced.

## 4. Conclusions

The PTFE and the MWCNTs were modified by adding an appropriate amount of silane coupling agent, and then the addition of the M-MWCNT-PTFE for the generation of the SCC was reduced. Subsequently, the WPU SCCs with M-MWCNT-PTFE were obtained using EsS. The chemical reaction that occurred between the coupling agent and the polar groups (such as -OH and -COOH) of the molecule of the WPU and on the surfaces of the PTFE and the MWCNTs caused the WPU, PTFE, and MWCNTs to become crosslinked together. Then, a relatively even coating was formed, leading to reduced curing interfaces between WPU and PTFE, and an improved corrosion resistance of the composite. In addition, the dehydration reaction that occurred between the silane coupling agent and the polar surface of Fe formed -NH_2_ groups, resulting in the enhanced adhesion of the coating and the solution of problems such as low adhesion, easy delamination, and wear. The properties of the WPU SCCs with M-MWCNT-PTFE were better than #0. With an increased filler content, the electrical conductivity and the hydrophobicity both increased, whilst the adhesion and the anticorrosive property first strengthened and then diminished. 

The structure and property of the composite with the WPU, PTFE, MWCNTs, and KH-550 at a mass ratio of 7:1.5:0.1:0.032 were the most excellent; its volume resistivity and WCA were 1.5 × 10^4^ (Ω·cm) and 155°, respectively. Its adhesion was increased by 7.9% and its corrosion current density was decreased by about three orders of magnitude compared with the pure WPU coating. This provides a foundation for the fabrication and the application of anticorrosive and conductive waterborne composites.

## Figures and Tables

**Figure 1 polymers-14-03093-f001:**
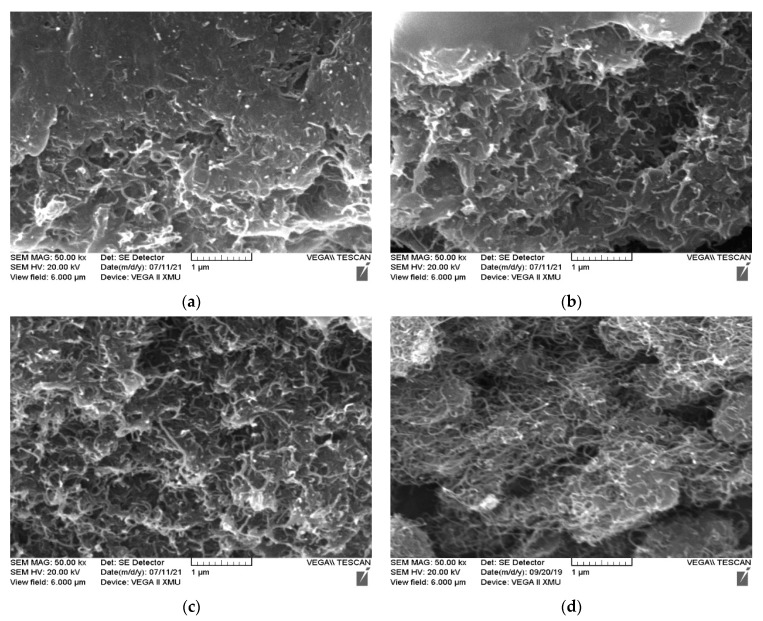
Surface micromorphologies of (**a**) 1#, (**b**) 2#, (**c**) 3#, and (**d**) 4# under a magnification of 50.00 k× and that of (**e**) 0# under a magnification of 2.00 k×.

**Figure 2 polymers-14-03093-f002:**
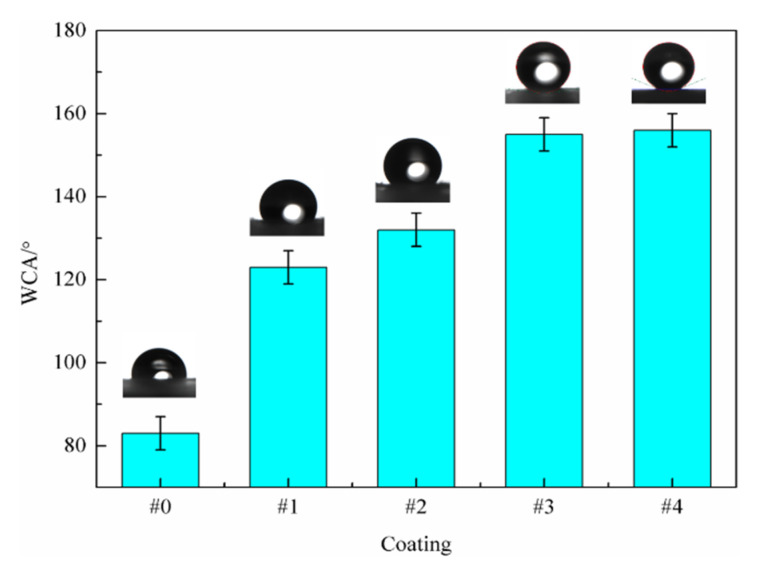
WCAs of different WPU SCCs with modified fillers.

**Figure 3 polymers-14-03093-f003:**
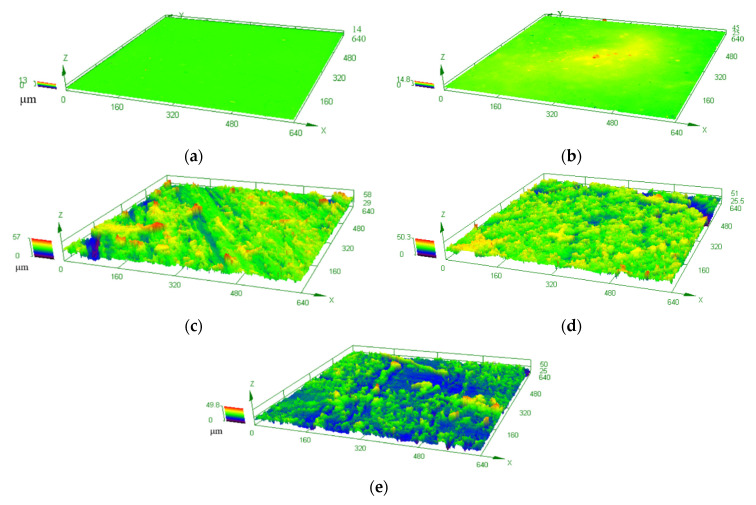
Surface 3D-micro morphologies of (**a**) 0#, (**b**) #1, (**c**) #2, (**d**) #3, and (**e**) #4.

**Figure 4 polymers-14-03093-f004:**
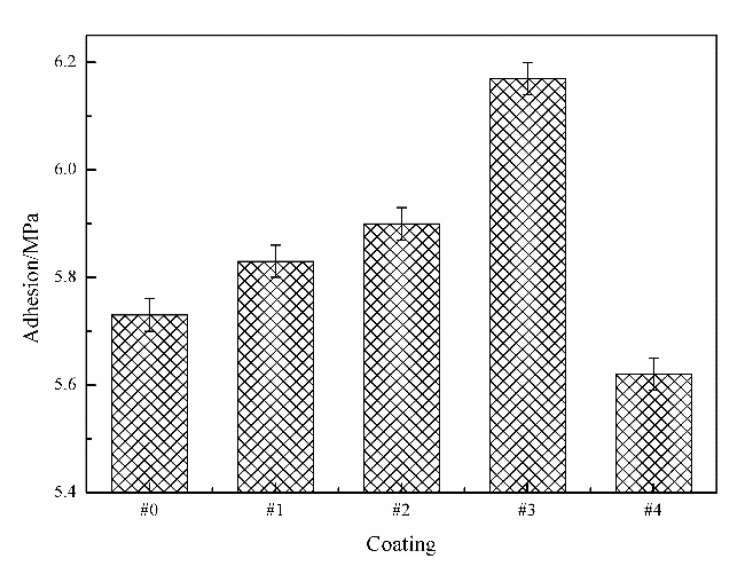
Adhesive property of different WPU SCCs with modified fillers.

**Figure 5 polymers-14-03093-f005:**
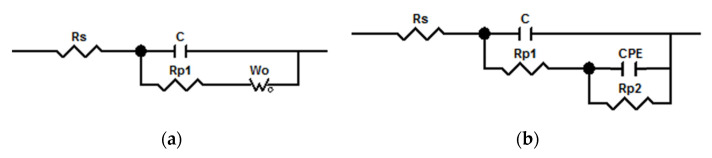
Equivalent electric circuit models with two time constants of (**a**) pure WPU coating and (**b**) WPU SCCs.

**Figure 6 polymers-14-03093-f006:**
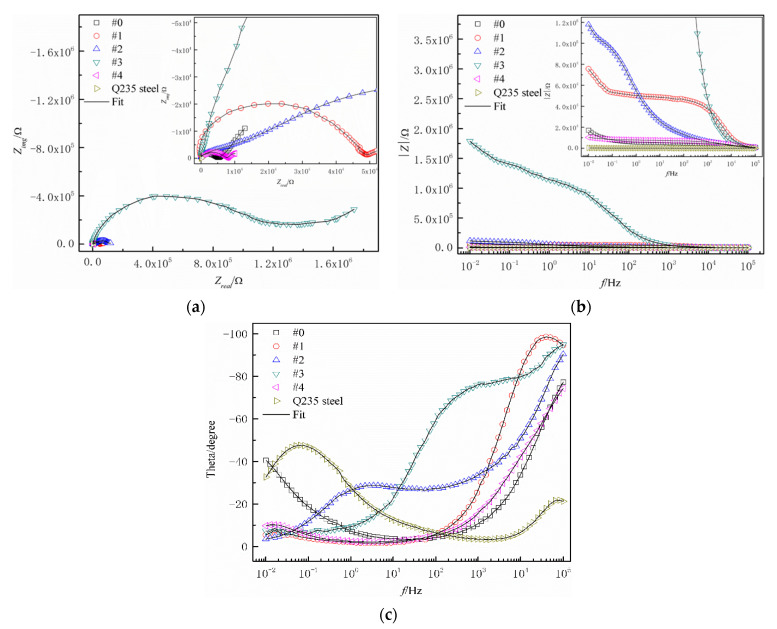
(**a**) Complex plane impedance spectra, (**b**) |Z|-*f*, and (**c**) θ-*f* for the WPU SCCs with modified fillers immersed in NaCl solution.

**Figure 7 polymers-14-03093-f007:**
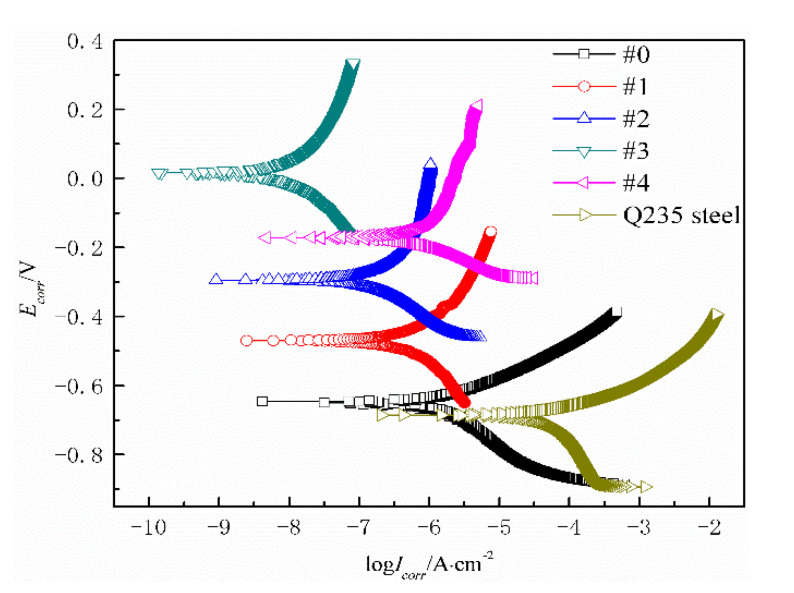
Potentiodynamic polarization curves of different WPU SCCs immersed in NaCl solution.

**Table 1 polymers-14-03093-t001:** Samples of the mixed WPU dispersions with modified fillers.

Dispersions	#1	#2	#3	#4
WPU/g	10	10	7	7
PTFE/g	1.5	2	1.5	2
MWCNTs/g	0.05	0.1	0.1	0.15
KH-550/g	0.031	0.042	0.032	0.043

**Table 2 polymers-14-03093-t002:** Process parameters of the sandblasting treatment.

Process Parameters	Spray Voltage (kV)	Compressed Air Pressure (MPa)	Feedwell Diameter (mm)	Liquid Flow Rate (mL·min^−1^)	Spray Distance (mm)	Spray Time (min)
	50~60	0.6~0.7	1	2	100~120	1~2

**Table 3 polymers-14-03093-t003:** Volume resistivities of different WPU SCCs with modified fillers.

Composites	#1	#2	#3	#4
Surface resistances (Ω)	(7.2 ± 0.5) × 10^6^	(4.2 ± 0.5) × 10^6^	(1.8 ± 0.5) × 10^6^	(1.1 ± 0.5) × 10^6^
Volume resistivities (Ω·cm)	(6.1 ± 0.5) × 10^4^	(3.5 ± 0.5) × 10^4^	(1.5 ± 0.5) × 10^4^	(9.3 ± 0.5) × 10^3^
Hardness (μm)	85.0 ± 4	84.5 ± 4	85.0 ± 4	84.5 ± 4

**Table 4 polymers-14-03093-t004:** Average values of surface roughness for different coatings.

Coatings	#0	#1	#2	#3	#4
Average surface roughness (μm)	0.137	0.160	3.113	3.339	3.674

**Table 5 polymers-14-03093-t005:** Fitted parameters of potentiodynamic polarization curves for the composites with different modified fillers.

Samples	#0	#1	#2	#3	#4	Q235 Steel
I/(A·cm^−2^)	1.3945 × 10^−6^	3.2091 × 10^−7^	4.8596 × 10^−7^	2.6172 × 10^−9^	1.0848 × 10^−6^	9.3593 × 10^−5^
E_vsSCE_/V	−0.6529	−0.4618	−0.2950	−0.0192	−0.3720	−0.6808
Corrosion rates/(mm·a^−1^)	0.01644	0.00378	0.00573	0.00003	0.01279	1.10370

## Data Availability

Data is contained within the article.

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
