# Peer review of "Properties and Fabrication of Waterborne Polyurethane Superhydrophobic Conductive Composites with Coupling Agent-Modified Fillers"

_polymers, 2022, doi:10.3390/polym14153093_

Round 1

Reviewer 1 Report

This article is comprehensive, logically organized, and contains valuable information on the properties and preparation of multiwalled carbon nanotubes (MWCNTs)/waterborne polyurethane (WUP) superhydrophobic conductive coatings with fillers modified by the coupling agent. The authors did excellent research on modifying the fillers including polytetrafluoroethylene (PTFE) emulsion and MWCNTs particles by a silane coupling agent to reduce their contents, afterward, preparing the MWCNTs/WPU superhydrophobic conductive coatings using electrostatic spraying. The authors demonstrated that the WPU, PTFE, and MWCNTs were linked into a whole by the coupling agent and then a relatively even coating was formed, leading to the reduced curing interfaces between WPU and PTFE and the improved corrosion resistance of the composite.

To improve the manuscript, the authors should take the following considerations:

(1) The authors presented the surface micromorphologies of (a) #1, (b) #3 and (c) #4 in Figure 1. It is suggested that the authors should present the surface micromorphologies of #0 and #2 as well for a better understanding of the subject.

(2) The authors presented the surface 3D-micromorphologies of (a) 0#, (b) #1, (c) #2, (d) #3 and (e) #4 in Figure 3. It is suggested that the authors should calculate and tabulate the surface roughness parameters to better understand the performance.

The submitted manuscript has significant scientific insights and the conclusions are soundly supported by the experimental data. However, the present submission requires minor revisions before being considered for publication in the Special Issue: Mechanical and Structure-Property Relationships of Polymer Composites in the esteemed Polymers in its current condition.

Reviewer 2 Report

This paper is a reasonable description of a new coating but needs a better description of chemistry behind how the coating was created.  In the abstract, I am unsure of the meaning of “contents” in “Fillers of Polytetrafluoroethylene (PTFE) and multiwalled carbon nanotubes (MWCNTs) were first treated by a coupling agent to reduce their contents.  I am assuming (at this point in the paper) that the impact of the filers on the compositive is minimized by the addition of the coupling agents, but the mechanism is not clear in the text.  It is unclear in the preparation if the coupling agent reacts with the filers, i.e. hydroxides or other function groups on the MWCNTs. I think a better description of the chemistry so the reader can understand. This addition could just be a few sentences.

There is this sentence (It was concluded that the WPU, PTFE and MWCNTs were linked into a whole by the coupling agent) is in the conclusions and some other comments in the text.  However, it is not really explained/demonstrated. Again, a description of the chemistry is needed.  Maybe stating that the assumption is that the filters are crossed linked due to that chemistry is better than just stating it is in the text?

This sentence is awkward “In the cassie–Baxterstate, in accordance with Cassie–Baxter equation,..” (line 316). Maybe just “Accordance to the Cassie–Baxter equation”?  There are some issues with this equation and predictions, but one knows what the authors used in their computations.

In Figure 5, Cpe should be shown as a capacitor and not a connector.

Round 2

Reviewer 2 Report

The authors made positive corrections.  I fell the paper is ready for publications.